# An Improved Unfolded Coprime Linear Array Design for DOA Estimation with No Phase Ambiguity

**DOI:** 10.3390/s24196205

**Published:** 2024-09-25

**Authors:** Pan Gong, Xiaofei Zhang

**Affiliations:** 1College of Electronic Information and Integrated Circuits, Nanjing Vocational University of Industry Technology, Nanjing 211106, China; 2College of Electronic and Information Engineering, Nanjing University of Aeronautics and Astronautics, Nanjing 210016, China; zhangxiaofei@nuaa.edu.cn

**Keywords:** direction of arrival (DOA) estimation, sparse array, improved unfolded coprime linear array (IUCLA), Cramer–Rao bound (CRB), multiple signal classification (MUSIC)

## Abstract

In this paper, the direction of arrival (DOA) estimation problem for the unfolded coprime linear array (UCLA) is researched. Existing common stacking subarray-based methods for the coprime array are invalid in the case of its subarrays, which have the same steering vectors of source angles. To solve the phase ambiguity problem, we reconstruct an improved unfolded coprime linear array (IUCLA) by rearranging the reference element of the prototype UCLA. Specifically, we design the multiple coprime inter pairs by introducing the third coprime integer, which can be pairwise with the other two integers. Then, the phase ambiguity problem can be solved via the multiple coprime property. Furthermore, we employ a spectral peak searching method that can exploit the whole aperture and full DOFs of the IUCLA to detect targets and achieve angle estimation. Meanwhile, the proposed method avoids extra processing in eliminating ambiguous angles, and reduces the computational complexity. Finally, the Cramer–Rao bound (CRB) and numerical simulations are provided to demonstrate the effectiveness and superiority of the proposed method.

## 1. Introduction

Direction of arrival (DOA) estimation has attracted much attention in recent years, and it has been widely applied in many fields, such as wireless communication, radar, sonar, medicine and other engineering applications [1,2,3,4,5,6,7,8]. Additionally, the recent integration of neural networks [9] into the domain of DOA marks a promising frontier in the landscape of next-generation wireless communications. Thus, a deep learning-based scheme is proposed, and the simulation results affirm the superior performance. The model’s robustness is rigorously examined across various validation cases, providing conclusive evidence of its potential in real-world applications. The accurate DOA estimation and low computational complexity have become increasingly important in recent years. In past decades, numerous subspace-based DOA estimation algorithms, e.g., multiple signals classification (MUSIC) [10,11,12,13,14,15,16,17] and estimation of signal parameters via rotational invariance techniques (ESPRIT) [18,19,20,21,22], have been proposed for uniform linear arrays (ULA) [23,24] and have attracted much attention due to their high resolution and performance. However, the adjacent antenna element spacing is required to be less than half of a wavelength to avoid the phase ambiguity problem [25]. Additionally, it limits the array aperture and consequently influences the mutual coupling [26] between the elements and the estimation performance; therefore, a non-uniform linear array has been recently proposed to increase the degrees of freedom (DOF) of the array, known as sparse arrays. Coprime arrays [27,28] have aroused wide attention due to the improvement of DOA estimation performance. Meanwhile, due to the larger array aperture, less mutual coupling effects arise. Furthermore, a coprime linear array (CLA) consists of two ULAs, with the inter-element spacing larger than half-wavelength; therefore, higher resolution, larger array aperture and a lesser mutual coupling effect can be attained [29,30]. Recently, a subarray-based method for a coprime array to solve the ambiguity problem was developed, and it is proposed in [31]. This algorithm considers the whole array as two subarrays, and it processes the two subarrays separately. Then, the MUSIC algorithm is applied to these two subarrays, respectively. By combining the estimates and finding the nearest spectral peaks from the MUSIC spectra of two subarrays, the phase ambiguity problem can be eliminated. However, this method suffers from severe computational complexity due to the global angular searching. Then, a partial spectral search method is proposed in [32] to decrease the computational complexity, which employs the linear relationship, along with ambiguous DOA estimates, and searches through a small sector. However, some problems occur in these mentioned methods of the CLA, which are as follows:DOF is limited by the subarray, which has a smaller number of elements;Only self-information of two subarrays is exploited, but mutual information is neglected; as a result, DOA estimation performance is degraded;These methods need extra procedures to eliminate ambiguous angles.

To solve these problems, an unfolded coprime linear array (UCLA) is proposed, which unfolds the two subarrays in two opposite directions so that the array aperture is extended [33]. By stacking the directional matrices, both self-information and mutual information are utilized. Therefore, the ambiguity problem is suppressed. Furthermore, this method can achieve the full DOFs due to the employment of the whole array sensors. Nevertheless, this technique is not always true. When the source signals satisfy some relations, the method will be invalid. In the case of three signals, for the two subarrays, when there are two different signals that have the same steering vectors as the given DOA, the phase ambiguity still exists. Aiming to tackle this problem, Yang et al. proposed a beamforming-based technique by defining a decision variable [34] to eliminate the phase ambiguity. However, this method employs other techniques, in addition to MUSIC spectral searching, to distinguish the real DOAs, which can increase the computational complexity. Meanwhile, this method does not always work, and it usually depends on the decision variable. When the decision variable is small, phase ambiguity problems still occur.

The method in [33] can achieve DOA estimation in most cases, but it does not work in special conditions. We have demonstrated that the method will be invalid when two subarrays have the same steering vectors of the source angles. The method in [34] can achieve the DOAs’ estimation; however, along with MUSIC, it needs additional techniques to eliminate the ambiguous angles, which will increase the computational complexity. We demonstrated that this method will sometimes have the problem of “false targets” when the decision variable is small. We can directly solve the problem of “false targets” by designing a multiple coprime array configuration. Furthermore, numerous simulation results are provided to verify the effectiveness of the constructed array. Moreover, compared to nested arrays, the physical aperture of coprime arrays is larger, and our designed array has a larger aperture than the original coprime array, which can achieve higher angular resolution.

Therefore, the MUSIC algorithm can be employed directly, without ambiguous angles arising. Meanwhile, we can obtain the DOAs’ estimation and do not need any other technique to estimate it. The method can achieve the full DOFs of the array. Compared to original methods, we find that the proposed method can detect sources effectively, without phase ambiguity problems.

The main contributions of this paper can be summarized as follows.

(1)The proposed method can effectively eliminate the phase ambiguity problem by rearranging the reference element spacing and determining unique directional vectors of different angles.(2)The whole array aperture of the designed array configuration is utilized to increase DOFs and improve DOA estimation performance.(3)The proposed method can obtain unambiguous DOA estimates without additional processing, which can reduce computational complexity.(4)CRB and numerical simulations are offered to verify the superiorities and effectiveness of the proposed method.

The paper is organized as follows: Section 2 introduces the array configuration and the data model of the received signals. In Section 3, we detail the ambiguity problem and present the proposed method; then, we analyze the DOFs and Cramer–Rao bound (CRB) of the proposed method. In Section 4, we provide extensive simulation results and demonstrate the superiority of the proposed method. Finally, we present the conclusions in Section 5.

## 2. Array Signal Model

As depicted in Figure 1, we consider an UCLA with $T=M_{1}+M_{2}-1$ sensors. For the array, it is composed of two uniform linear subarrays. One subarray has $M_{1}$ sensors and the other one has $M_{2}$ sensors, respectively, and $M_{1}$ and $M_{2}$ are a pair of coprime integers. Meanwhile, we assume $M_{1}<M_{2}$. Two subarrays are overlapping at the position of (0,0), so we can compute the total sensors of the array as $T=M_{1}+M_{2}-1$. In Figure 1, the inter-element spacing of subarray 1 can be denoted as $d_{1}=M_{2}d=M_{2}\lambda /2$, and the inter-element spacing of subarray 2 can be denoted as $d_{2}=M_{1}d=M_{1}\lambda /2$, where d=λ/2 and λ represents the wavelength.

Assume that there are *K* narrowband sources impinging on the array, which locates at $\Theta =[\theta_{1},\theta_{2},\cdots ,\theta_{K}]$, with signal powers of $\sigma_{1}^{2},\sigma_{2}^{2},\;\cdots ,\sigma_{K}^{2}$. Additionally, $\theta_{k}\in (-\pi /2,\pi /2)$ denotes the *k*-th signal, wherein K<T and k∈[1,2,⋯,K]. Furthermore, we suppose that these signals are far-field uncorrelated. The received signal at time t is denoted as follows [10]:(1)$$x\left(t\right)=\left[\begin{array}{c} x_{1}(t)\\ x_{2}(t) \end{array}\right]=\left[\begin{array}{c} A_{1}\\ A_{2} \end{array}\right]s\left(t\right)+\left[\begin{array}{c} n_{1}(t)\\ n_{2}(t) \end{array}\right]\;\;\;\;\;\;\;=As(t)+n(t)$$
where $A=[A_{1}^{T},A_{2}^{T}{]}^{T}$. $A_{1}$ and $A_{2}$ are steering matrices of the two subarrays, respectively. $A_{1}=[a_{1}(\theta_{1}),a_{1}(\theta_{2}),\cdots ,a_{1}(\theta_{K})]$ and $A_{2}=[a_{2}(\theta_{1}),a_{2}(\theta_{2}),\cdots ,a_{2}(\theta_{K})]$. $a_{1}(\theta_{k})=[1,$ $e^{jM_{2}\pi \sin\theta_{k}},\cdots ,e^{j\left(M_{1}-1\right)M_{2}\pi \sin\theta_{k}}{]}^{T}$ is the steering vector for subarray 1, and $a_{2}(\theta_{k})=$ $[e^{-j\left(M_{2}-1\right)M_{1}\pi \sin\theta_{k}},e^{-j\left(M_{2}-2\right)M_{1}\pi \sin\theta_{k}},\cdots ,e^{-jM_{1}\pi \sin\theta_{k}},1{]}^{T}$ is the steering vector for subarray 2, respectively. $s(t)=[s_{1}(t),$$s_{2}(t),\cdots ,s_{K}(t){]}^{T}$ is the signal emitted by the k-th target at time t, where t=1, 2,⋯,L and L represents the sampling number. n(t) stands for the additive white Gaussian noise, it obeys the normal distribution of $N(0,\sigma_{k}^{2})$ and it is independent from source signals. $n_{1}(t)$ is the noise vector of subarray 1, and $n_{2}(t)$ denotes the noise vectors of subarray 2.

## 3. Ambiguity Problem Demonstration and Resolution

This section introduces the original coprime linear array and the cause of the phase ambiguity problem. In addition, we analyze the existing methods.

### 3.1. Preview

Generally, to avoid phase ambiguity problems, the inter-element spacing is usually set to be no larger than half-wavelength; however, in this way, the array aperture is restricted. For the coprime array, which is made up of two uniform linear arrays (ULA), the inter-element spacing is larger than half of a wavelength.

Figure 2 depicts the relationship between the element spacing and the number of spectral peaks. Here, peaks denote the number of DOAs. Furthermore, we consider that there is only one signal coming from $\theta_{1}=24{}^{\circ}$. Figure 2, demonstrates that when the element spacing is set to be d=λ/2,  wherein λ represents the wavelength, no ambiguous angle appears. Meanwhile, when d=3λ/2 or d=5λ/2, ambiguous angles appear.

#### 3.1.1. Case 1

In Case 1, we consider that there are two signals coming from $\Theta =[\theta_{1},\theta_{2}]$.

We assume that there are two signals, $\theta_{1}$ and $\theta_{2}$, which are real angles. Then, we can obtain their corresponding directional vectors, which are $a(\theta_{1})$ and $a(\theta_{2})$.
(2)$$a(\theta_{1})={\left[a_{1}^{T}(\theta_{1}),a_{2}^{T}(\theta_{1})\right]}^{T}$$
(3)$$a(\theta_{2})={\left[a_{1}^{T}(\theta_{2}),a_{2}^{T}(\theta_{2})\right]}^{T}$$

We assume that $\theta_{1}^{\prime }$ and $\theta_{2}^{\prime }$ are ambiguous angles of $\theta_{1}$ and $\theta_{2}$. Therefore, we can acquire the corresponding steering vectors, which are $a(\theta_{1}^{\prime })$ and $a(\theta_{2}^{\prime })$.
(4)$$a(\theta_{1}^{\prime })=a(\theta_{1})={\left[a_{1}^{T}(\theta_{1}^{\prime }),a_{2}^{T}(\theta_{1}^{\prime })\right]}^{T}={\left[a_{1}^{T}(\theta_{1}),a_{2}^{T}(\theta_{1})\right]}^{T}$$
(5)$$a(\theta_{2}^{\prime })=a(\theta_{2})={\left[a_{1}^{T}(\theta_{2}^{\prime }),a_{2}^{T}(\theta_{2}^{\prime })\right]}^{T}={\left[a_{1}^{T}(\theta_{2}),a_{2}^{T}(\theta_{2})\right]}^{T}$$

Correspondingly, we have
(6)$$\left\{\begin{array}{c} a_{1}(\theta_{1})=a_{1}(\theta_{1}^{\prime })\\ a_{2}(\theta_{1})=a_{2}(\theta_{1}^{\prime })\\ a_{1}(\theta_{2})=a_{1}(\theta_{2}^{\prime })\\ a_{2}(\theta_{2})=a_{2}(\theta_{2}^{\prime }) \end{array}\right.$$

Then we have
(7)$$\left\{\begin{array}{c} M_{2}\pi \sin\theta_{1}=M_{2}\pi {\sin\theta }_{1}^{\prime }+2k_{1}\pi \\ M_{1}\pi \sin\theta_{1}=M_{1}\pi {\sin\theta }_{1}^{\prime }+2k_{2}\pi \\ M_{2}\pi \sin\theta_{2}=M_{2}\pi {\sin\theta }_{2}^{\prime }+2k_{1}\pi \\ M_{1}\pi \sin\theta_{2}=M_{1}\pi {\sin\theta }_{2}^{\prime }+2k_{2}\pi \end{array}\right.$$
where $k_{1}=-(M_{2}-1),\cdots ,(M_{2}-1)$ and $k_{2}=-(M_{1}-1),$ $\cdots ,(M_{1}-1)$.

Then we can obtain
(8)$$\left\{\begin{array}{c} \sin\theta_{1}={\sin\theta }_{1}^{\prime }+2k_{1}/M_{2}\\ \sin\theta_{1}={\sin\theta }_{1}^{\prime }+2k_{2}/M_{1}\\ \sin\theta_{2}={\sin\theta }_{2}^{\prime }+2k_{1}/M_{2}\\ \sin\theta_{2}={\sin\theta }_{2}^{\prime }+2k_{2}/M_{1} \end{array}\right.$$

Then we have
(9)$$\frac{2k_{1}}{M_{2}}=\frac{2k_{2}}{M_{1}}$$

Due to the coprime property of $M_{1}$ and $M_{2}$, Equation (9) is not satisfied. That is, in the case of the two signals of $\theta_{1}$ and $\theta_{2}$, a phase ambiguity problem does not occur.

Figure 3 depicts spectrums without ambiguity, using the method proposed in [33]. The two signals are $\theta_{1}=10{}^{\circ},\theta_{2}=37{}^{\circ}$. SNR and the snapshots are set to be SNR=5dB and L=200, respectively. Therefore, we can draw a conclusion that the method proposed in [33] can tackle the phase ambiguity problem with Case 1.

#### 3.1.2. Case 2

In Case 2, we consider three signals with $\Theta =[\theta_{1},\theta_{2},\theta_{3}]$.

Similar to Case 1, we suppose $\theta_{1}^{\prime }$ and $\theta_{2}^{\prime }$ to be ambiguous angles of $\theta_{1}$ and $\theta_{2}$, respectively. Then, we can obtain the corresponding directional vectors of $a(\theta_{1}^{\prime })$ and $a(\theta_{2}^{\prime })$. If the relationship that the sine function of the third signal $\theta_{3}$ equals to $a(\theta_{1}^{\prime })$ and $a(\theta_{2}^{\prime })$ is satisfied, the phase ambiguity problem appears. Figure 4 shows the spectrums with the method in [33], where it has three target signals, which are $\theta_{1}=10{}^{\circ},\theta_{2}=20{}^{\circ}$ and $\theta_{3}=30{}^{\circ}.$ We set the number of snapshots to be L=200 and SNR=5dB. It is depicted clearly in Figure 4 that the method proposed in [33] still has no difficulty to resolve the three source signals. However, this method is not always accurate. When these three source signals satisfy the relationship wherein the sine function of the third signal, $\theta_{3}$, equals $a(\theta_{1}^{\prime })$ and $a(\theta_{2}^{\prime })$, the method in [33] will not work; we will illustrate this in Case 3.

#### 3.1.3. Case 3

From Case 2, we notice that the method in [33] can effectively detect three signals without ambiguous angles, despite the fact that it does not always work. Thus, we illustrate Case 3 as follows.

We consider three target signals, which come from $\Theta =[\theta_{1},\theta_{2},\theta_{3}]$. Similar to Case 2, we suppose $\theta_{1}^{\prime }$ and $\theta_{2}^{\prime }$ to be ambiguous angles of $\theta_{1}$ and $\theta_{2}$, respectively. Then, we can obtain the corresponding directional vectors of $a(\theta_{1}^{\prime })$ and $a(\theta_{2}^{\prime })$. When these three signals satisfy that the sine function of $\theta_{3}$ equals to $a(\theta_{1}^{\prime })$ and $a(\theta_{2}^{\prime })$, the ambiguous angle will arise. In other words, a phase ambiguity problem appears, which is invalidated in the following.

We suppose that the sine function of the third angle, $\theta_{3}$, equals to $a(\theta_{1}^{\prime })$ and $a(\theta_{2}^{\prime })$. Then, it has
(10)$$\left\{\begin{array}{c} \sin\theta_{3}=\sin\theta_{1}+2(-k_{1})/M_{2}\\ \sin\theta_{3}=\sin\theta_{2}+2(-k_{2})/M_{1} \end{array}\right.$$

Then, we can obtain the relationship among $\theta_{1}$,$\theta_{2}$ and $\theta_{3}$ as follows:(11)$$\left\{\begin{array}{c} M_{2}\pi \sin\theta_{3}=M_{2}\pi \sin\theta_{1}+2(-k_{1})\pi \\ M_{1}\pi \sin\theta_{3}=M_{1}\pi \sin\theta_{2}+2(-k_{2})\pi \end{array}\right.$$
where $k_{1}=-(M_{2}-1),\cdots ,-1,1,\cdots ,(M_{2}-1)$ and $k_{2}=-(M_{1}-1),\cdots ,-1,1,\cdots ,(M_{1}-1)$.

It has
(12)$$\left\{\begin{array}{c} a_{1}(\theta_{3})=a_{1}(\theta_{1})\\ a_{2}(\theta_{3})=a_{2}(\theta_{2}) \end{array}\right.$$
so we have
(13)$$\left\{\begin{array}{c} a_{1}(\theta_{1})+a_{1}(\theta_{2})-a_{1}(\theta_{3})=a_{1}(\theta_{2})\\ a_{2}(\theta_{1})+a_{2}(\theta_{2})-a_{2}(\theta_{3})=a_{2}(\theta_{1}) \end{array}\right.$$

Then we define
(14)$$\left\{\begin{array}{c} a_{1}(\theta_{4})=a_{1}(\theta_{1})+a_{1}(\theta_{2})-a_{1}(\theta_{3})\\ a_{2}(\theta_{4})=a_{2}(\theta_{1})+a_{2}(\theta_{2})-a_{2}(\theta_{3}) \end{array}\right.$$

It has
(15)$$\left\{\begin{array}{c} a_{1}(\theta_{4})=a_{1}(\theta_{2})\\ a_{2}(\theta_{4})=a_{2}(\theta_{1}) \end{array}\right.$$

It is obvious that the forth angle, $\theta_{4}$, arises. That is to say, a phase ambiguity problem occurs. Simulations are presented to demonstrate the analysis.

Figure 5 depicts the scenery of three signals, $\theta_{1}=12.37{}^{\circ},$ $\theta_{2}=30{}^{\circ}$ and $\theta_{3}=64.16{}^{\circ}$, that come to the array. The SNR and the number of snapshots are set to be SNR=5 dB and L=200, respectively. It can be found that these three signals satisfy $a_{1}(\theta_{3})=a_{1}(\theta_{1})$ and $a_{2}(\theta_{3})=a_{2}(\theta_{2})$. Furthermore, we have demonstrated that the method in [33] is not effective in Case 2. Figure 5 shows that, other than these three signals, there is a forth spectrum that was detected. Aiming at solving the ambiguity problem, Yang et al. proposed a modified method by defining a decision variable [34]. Meanwhile, the method combines the beamforming technique with MUSIC, but it is not always effective. Figure 6 shows that five signals come to the array, and, by using the method in [34], the five signals can be detected successfully; however, they still have the ambiguous angle, which can increase the ineffectiveness of the DOA estimation performance. To tackle this problem, an improved unfolded coprime linear array (IUCLA) for the DOA estimation was proposed without the phase ambiguity problem. In the proposed method, we choose the third coprime integer, and, by moving the reference element, the linear combination relation of the steering vectors can be eliminated. As a result, we can use spectrum peak searching to achieve the real DOAs. Simultaneously, no additional algorithm is needed. The method is illustrated in the following part.

## 4. Proposed Method and Theoretical Performance Analysis

### 4.1. The Proposed Method

This section presents the proposed method to tackle the phase ambiguity problem with Case 3. In the proposed method, we consider constructing the multiple coprime arrays by introducing the third coprime integer. Specifically, Figure 7a presents an UCLA, which is composed of two subarrays, including subarray 1 with $M_{1}$ sensors and subarray 2 with $M_{2}$ sensors. Figure 7b introduces the third coprime integer, $M_{3}$, and presents a rearranged reference point instead of (0, 0). For the improved array, it still incorporates two subarrays which include $M_{1}$ and $M_{2}$ sensors, respectively. Furthermore, we can notice that the reference sensor is rearranged from (0, 0) to $\left(M_{3}d,0\right)$, where $M_{3}$ can make coprime pairs with both *M* and *N*.

We employ the third coprime number, and it combines with another coprime number to form a pair of coprime integers with pairwise coprime. Multiple pairwise coprime integers are employed and the strong sidelobes can be suppressed by coprime property. Similar to the UCLA in Section 1, aside from two coprime integers, $M_{1}=5,M_{2}=7$, we employ the third coprime integers, which are $M_{3}=3$. In this way, there is not only one coprime integers pair, but also another two coprime integer pairs. Similar to our prior work, we multiply and employ the coprime property, and can eliminate the strong sidelobe.

In the rearranged array, we can denote the directional vectors of two subarrays with the *k*-th signal as
(16)$$a_{11}(\theta_{k})={\left[e^{j3M_{2}\pi \sin\theta_{k}},e^{jM_{2}\pi \sin\theta_{k}},\cdots ,e^{j(M_{1}-1)M_{2}\pi \sin\theta_{k}}\right]}^{T}$$
(17)$$a_{22}(\theta_{k})={\left[e^{-j(M_{2}-1)M_{1}\pi \sin\theta_{k}},e^{-j(M_{2}-2)M_{1}\pi \sin\theta_{k}},\cdots ,e^{-jM_{1}\pi \sin\theta_{k}},e^{-j3M_{2}\pi \sin\theta_{k}}\right]}^{T}$$

Then, we can obtain the steering vectors of source signal $\theta_{3}$
(18)$$a_{11}(\theta_{3})=[e^{j3M_{2}\pi \sin\theta_{3}},e^{jM_{2}\pi \sin\theta_{3}},\;\cdots ,e^{j(M_{1}-1)M_{2}\pi \sin\theta_{3}}{]}^{T}$$
(19)$$a_{22}(\theta_{3})=[e^{-j\left(M_{2}-1\right)M_{1}\pi \sin\theta_{3}},e^{-j\left(M_{2}-2\right)M_{1}\pi \sin\theta_{3}},\;\cdots ,e^{-jM_{1}\pi \sin\theta_{3}},e^{-j3M_{2}\pi \sin\theta_{3}}{]}^{T}$$

Additionally, $a(\theta_{k})$ denotes the directional vector of the total array.
(20)$$a(\theta_{k})={\left[a_{11}^{T}(\theta_{k}),a_{22}^{T}(\theta_{k})\right]}^{T}$$

Similar to Equation (1), we can obtain the received signal
(21)$$x_{iu}\left(t\right)=\left[\begin{array}{c} x_{iu1}(t)\\ x_{iu2}(t) \end{array}\right]=\left[\begin{array}{c} A_{11}\\ A_{22} \end{array}\right]s\left(t\right)+\left[\begin{array}{c} n_{11}(t)\\ n_{22}(t) \end{array}\right]\;\;\;\;=A_{iu}s(t)+n_{iu}(t)\;$$
where $A_{iu}=[A_{11}^{T},A_{22}^{T}{]}^{T}$. $A_{11}$ and $A_{22}$ are steering matrices of the two subarrays for IUCLA, respectively. $A_{11}=[a_{11}(\theta_{1}),a_{11}(\theta_{2}),\cdots ,a_{11}(\theta_{K})]$ and $A_{22}=[a_{22}(\theta_{1}),a_{22}(\theta_{2}),\cdots ,$ $a_{22}(\theta_{K})],$ where $a_{11}(\theta_{k})$ and $a_{22}(\theta_{k})$ are denoted as Equations (16) and (17). $n_{iu}(t)=[n_{11}^{T},n_{22}^{T}{]}^{T}$ is the total array noise vector.

The corresponding total covariance matrix can be computed with L snapshots
(22)$${\hat{R}}_{iu}=(1/L)\sum_{l=1}^{L}X_{iu}X_{iu}^{H}$$

The eigenvalue decomposition result of the total covariance matrix, ${\hat{R}}_{iu}$, can be denoted as
(23)$${\hat{R}}_{iu}={\hat{E}}_{siu}{\hat{D}}_{siu}{\hat{E}}_{siu}^{H}+{\hat{E}}_{niu}{\hat{D}}_{niu}{\hat{E}}_{niu}^{H}$$
where ${\hat{E}}_{siu}$ and ${\hat{E}}_{niu}$ are the signal subspace and noise subspace matrices, respectively, and ${\hat{D}}_{siu}$ and ${\hat{D}}_{niu}$ include the eigenvalues.

Referring to the orthogonality between the signal subspace and the noise subspace, the spectral peak function of MUSIC can be denoted as [10]
(24)$$f(\theta )=\frac{1}{a_{iu}^{H}(\theta )E_{niu}E_{niu}^{H}a_{iu}(\theta )}$$
where $a_{iu}(\theta )={\left[a_{11}^{T}(\theta ),a_{22}^{T}(\theta )\right]}^{T}$.

Referring to the derivation above, when there are three signals coming to the array and they satisfy $a_{11}(\theta_{3})=a_{11}(\theta_{1})$ and $a_{22}(\theta_{3})=a_{22}(\theta_{2})$, a phase ambiguity problem arises. In the following, we focus on proving and resolving the ambiguity problem.

**Proof.** Assume $a_{11}(\theta_{3})=a_{11}(\theta_{1})$. $a_{11}(\theta_{3})$ and $a_{11}(\theta_{1})$ represent the steering vectors of $\theta_{3}$ and $\theta_{1}$ with subarray 1, respectively. It has
(25)$$\left\{\begin{array}{l} 3M_{2}\pi \sin\theta_{3}=3M_{2}\pi \sin\theta_{1}+2k_{1}\pi \\ M_{2}\pi \sin\theta_{3}=M_{2}\pi \sin\theta_{1}+2k_{1}\pi \end{array}\right.$$
where $k_{1}=(-M_{2},M_{2})$. Thus, $a_{11}(\theta_{3})\neq a_{11}(\theta_{1})$. □

Similarly, we can obtain that $a_{22}(\theta_{3})\neq a_{22}(\theta_{2})$, where $a_{22}(\theta_{3})$ and $a_{22}(\theta_{1})$ represent the steering vectors of $\theta_{3}$ and $\theta_{1}$ with subarray 2, respectively. Furthermore, we can reveal that the spectral peak function is eliminated. In this way, we can obtain the accurate DOAs’ estimation without ambiguous angle $\theta_{4}$, which means that the phase ambiguity problem is solved.

### 4.2. Theoretical Performance Analysis

#### 4.2.1. DOF Analysis

In this part, we will provide the DOF performance of the proposed method. The method can achieve the full DOFs. Figure 8 shows that there are three signals coming to the array, with $M_{1}=3,M_{2}=2$. Furthermore, the signals are $\theta_{1}=10{}^{\circ},\theta_{2}=27.35{}^{\circ}$ and $\theta_{3}=35.01{}^{\circ}$.

Figure 9 shows that there are seven signals coming to the array, with $M_{1}=5,\;M_{2}=4$. Furthermore, the signals are denoted as $\theta_{1}=-30{}^{\circ},\;\theta_{2}=-10{}^{\circ},\;\theta_{3}=10{}^{\circ},\;\theta_{4}=30{}^{\circ},\;\theta_{5}=35{}^{\circ},\;\theta_{6}=40{}^{\circ}$ and $\theta_{7}=50{}^{\circ}$. From Figure 8 and Figure 9, we can see that the proposed method can achieve the full DOF.

#### 4.2.2. Computational Complexity Analysis

We compute the complexity of the proposed method and compare the complexity with the methods in [33,34]. The complexity of the proposed method is similar to the method in [33], which is denoted as $O(T^{2}L+T^{3}+GT(T-K))$, where $T=M_{1}+M_{2}-1$. Furthermore, G=180°/τ is the number of the spectrum searching, where τ is the searching step, and we usually set τ=0.1°. L is the number of the snapshots. The method in [34] needs an additional algorithm to distinguish the true DOAs from the MUSIC spectrum. Thus, the complexity exceeds the method in [33] and the proposed method, which is $O(T^{2}L+T^{3}+GT(T-K)+QT^{2})$. Q is the number of searching of the additional beamforming technique. Table 1 presents the computational complexity comparison and the running time of the methods above, which are computed using MATLAB R2015b under the condition of Intel (R) Xeon (R) CPU E430 @3.10 GHz and 8 GB random access memory, where $L=200,K=3,(\theta_{1},\phi_{1})=(20{}^{\circ},38.88{}^{\circ},47.90{}^{\circ}),\;$ $M_{1}=5,M_{2}=4,$ which clearly shows that the proposed method is similar to the method in [33] and outperforms the method in [34]. Figure 10 depicts the complexity comparison versus the number of sensors with subarray 2, where the number of subarray 1 is $M_{1}=5$. As the proposed method does not need an additional procedure to identify the targets, it shows clearly that its complexity is much lower than the method in [34], and is close to the method in [33].

#### 4.2.3. Advantages

In this part, we summarize the advantages of the improved array of IUCLA for DOA estimation.(1)The proposed method can effectively eliminate the phase ambiguity problem by rearranging the reference element spacing and breaking the spectral function with the directional matrix.(2)The improved array treats the array as a whole so that it can achieve the full DOFs; hence, it improves the estimation performance. Nevertheless, the method in [31] has a great loss in DOF, which can acquire, at most, $M_{1}-1$ DOFs, where we assume $M_{1}<M_{2}$.(3)CRB, as the lower bound for the unbiased estimation, is provided as a standard to measure the estimation performance. Furthermore, it is proven that the designed array can achieve the lower CRB.(4)The designed array can achieve DOA estimation with an excellent DOA estimation performance compared to other algorithms, and it does not need additional algorithms.(5)The proposed method performs a significant complexity decrease compared to the method in [33] due to the fact that the method does not use an additional technique, and it has a slight degrading of estimation performance compared to the method in [34].


#### 4.2.4. Cramer–Rao Bound

CRB is the lower bound for unbiased estimation. We provide the CRB as a standard to measure the estimation performance [35,36]. In this section, we derive the CRB of the designed IUCLA.

First, we construct the steering matrix of IUCLA as
(26)$$A_{\alpha }=\left[\begin{array}{c} \;A_{11}\\ \;A_{\chi } \end{array}\right]$$
where $A_{\chi }$ represents a sub-matrix containing the second row to the last one of $A_{22}$, since these two subarrays of IUCLA share the same element at the original point.

Referring to [36], we can obtain the CRB as
(27)CRB=σn22LRe⁡DH[I−Aα(AαHAα)−1AαH]D⊕Rs−1
where $R_{s}=\frac{1}{L}\sum_{t=1}^{L}s(t)s^{H}\left(t\right)$, $D=\left[\frac{\partial a_{\alpha ,1}}{\partial \theta_{1}},\frac{\partial a_{\alpha ,2}}{\partial \theta_{2}},\ldots ,\frac{\partial a_{\alpha ,K}}{\partial \theta_{K}}\right]$ and $a_{\alpha ,k}$ denotes the *k*-th column of $A_{\alpha }$.

## 5. Simulation Results and Discussion

In the simulation section, we validate the reliability of the proposed method compared to the methods in [33,34], where we employ an UCLA. Subarray 1 is with $M_{1}=5$ sensors and subarray 2 is with $M_{2}=7$ sensors. Furthermore, we present the comparison of different methods and arrays, including UCLA, CLA and the designed IUCLA.

### 5.1. Reliability Comparison Analysis Based on Different Methods

Example 1: We assume that three signals, $\theta_{1}=12.37{}^{\circ},\theta_{2}=30{}^{\circ}$ and $\theta_{3}=64.16{}^{\circ}$, will be coming to the array. It can be noticed that these signals satisfy the Equation (10). In the simulation, we set SNR = −5 dB and the number of snapshots to L=200. We provide the simulation result of the method in [33] using the proposed method. From Figure 11a, we can see that using the designed method has no ambiguity when three sources satisfy Equation (10), whereas the ambiguity-free method still has an ambiguity problem. In conclusion, the proposed method can tackle the phase ambiguity problem in Case 2.

Example 2: We assume three signals with angles of $\theta_{1}=20{}^{\circ},\theta_{2}=38.88{}^{\circ}$ and $\theta_{3}=47.90{}^{\circ}$. It can be noticed that these signals satisfy Equation (10). In the simulation, we set SNR = −5 dB and the number of snapshots to L=200. We provide the simulation results of the method in [34], using the proposed method. From Figure 11b, we can see that the method in [34] is not that effective, and the designed array can effectively solve the phase ambiguity problem.

### 5.2. Estimation Properties Analysis of the Proposed Method

For the same simulation scenario mentioned above, we utilize the Root Mean Square Error (RMSE) to validate the estimation accuracy of the proposed method, which is defined as [33]
(28)$$RMSE=\sqrt{\sum_{p=1}^{Q}\sum_{k=1}^{K}{\left({\hat{\theta }}_{k,p}-\theta_{k}\right)}^{2}/PK}$$
where P is the number of Monte Carlo simulations and ${\hat{\theta }}_{k,p}$ stands for the estimate of the p-th trial for the *k*-th theoretical angle $\theta_{k}$. In the rest of our paper, we set P=1000. Figure 12 and Figure 13 show the RMSE versus the SNR and the number of snapshots, respectively. The CRB is provided. It is observed that the proposed method improves with the increase in SNR and the number of snapshots, owing to its robustness against noise. Compared to ESPRIT and PM algorithms, it is obvious that the estimation accuracy of the proposed method performs the superior estimation behavior.

### 5.3. CRB Comparison Analysis Based on Different Arrays

We compare the estimation performance of several array configurations, including UCLA, conventional ULA and our designed IUCLA. In the simulation, for fair comparison, we assume that all arrays have the same number of sensors. We can conclude from Figure 14 and Figure 15 that the CRB and estimation performance of the proposed method are superior to the other two arrays.

## 6. Conclusions

In this paper, an IUCLA was proposed to address the ambiguity problem of the coprime array by rearranging the reference sensor to UCLA to reshape the directional vectors of subarrays. Subsequently, we introduce the third coprime integer. It combines another coprime number to form a pair of coprime integers with pairwise coprime. Multiple pairwise coprime integers are employed, and the strong sidelobes can be suppressed by coprime property. In this process, we do not need any other technique to eliminate the ambiguous problem. The proposed method not only improves the DOA estimation performance and the number of DOFs, but it also reduces the computational complexity. CRB analysis and extensive simulation results have demonstrated the superiorities and effectiveness of our method.

## Figures and Tables

**Figure 1 sensors-24-06205-f001:**
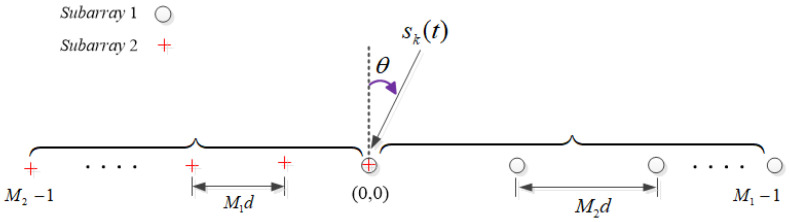
Unfolded coprime linear array (UCLA).

**Figure 2 sensors-24-06205-f002:**
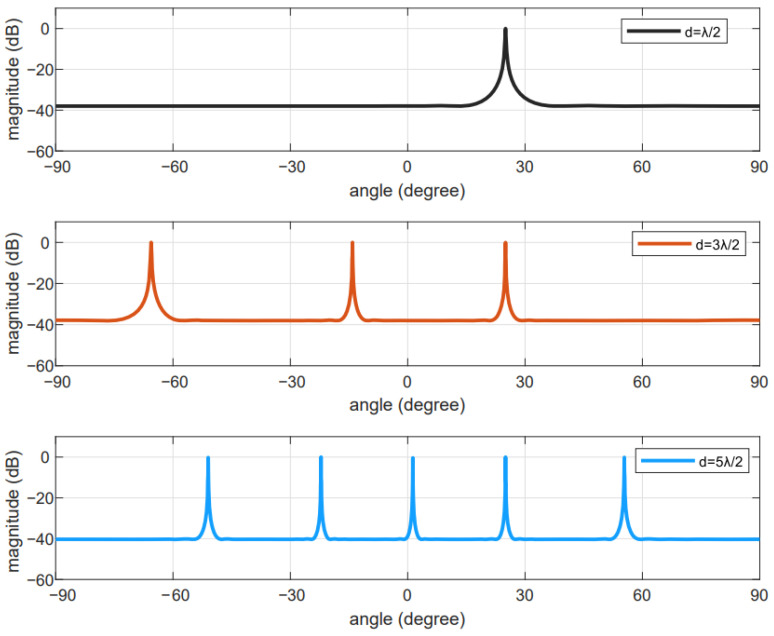
The relationship between the phase ambiguity problem and the inter-element spacing.

**Figure 3 sensors-24-06205-f003:**
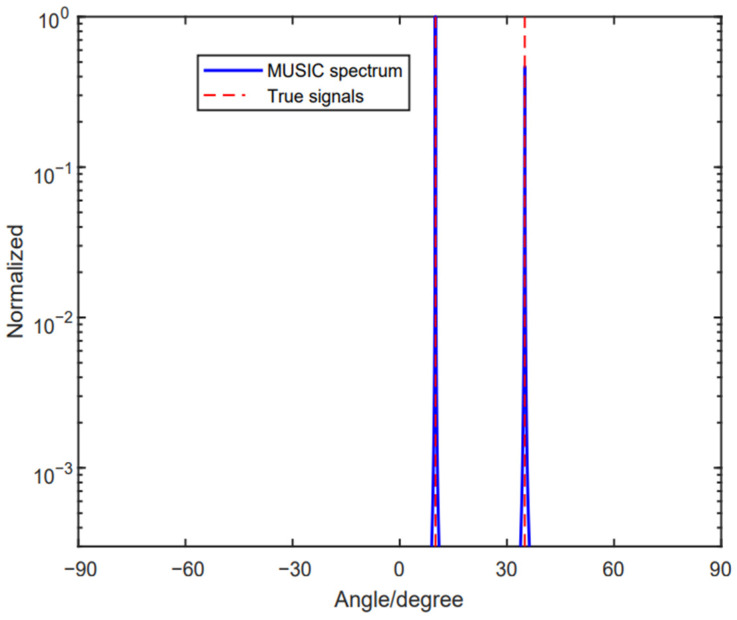
No ambiguous angle arises with two source signals, where $\theta_{1}=10{}^{\circ},\theta_{2}=37{}^{\circ}$.

**Figure 4 sensors-24-06205-f004:**
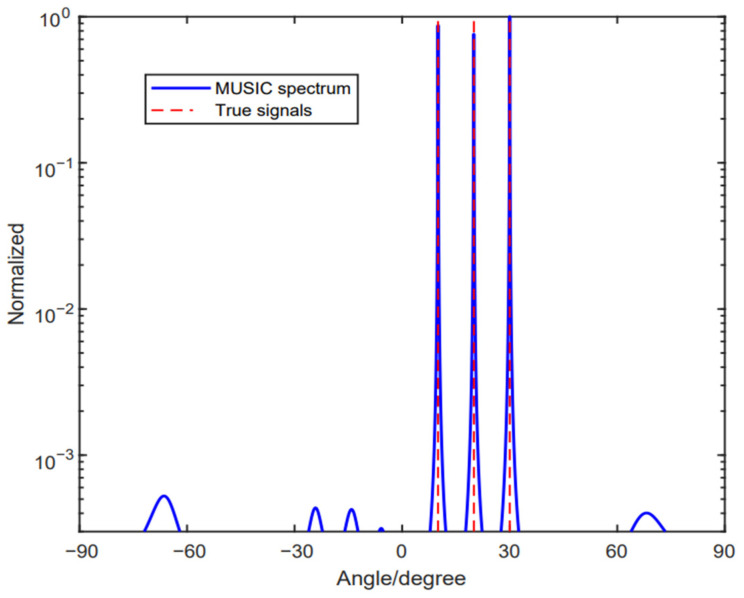
No ambiguous angle arises with the three given source signals, where $\theta_{1}=10{}^{\circ},\;\theta_{2}=20{}^{\circ},\;\theta_{3}=30{}^{\circ}$.

**Figure 5 sensors-24-06205-f005:**
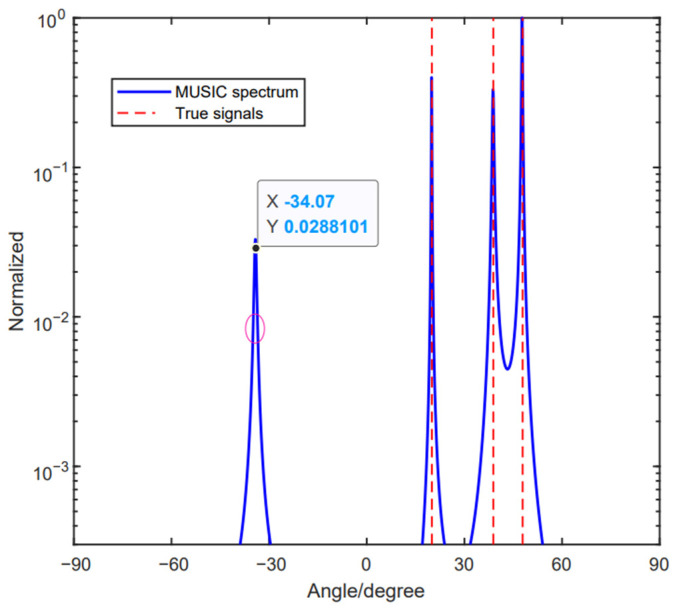
With the method in [33], the ambiguous angle arises with three source signals that satisfy Equation (10).

**Figure 6 sensors-24-06205-f006:**
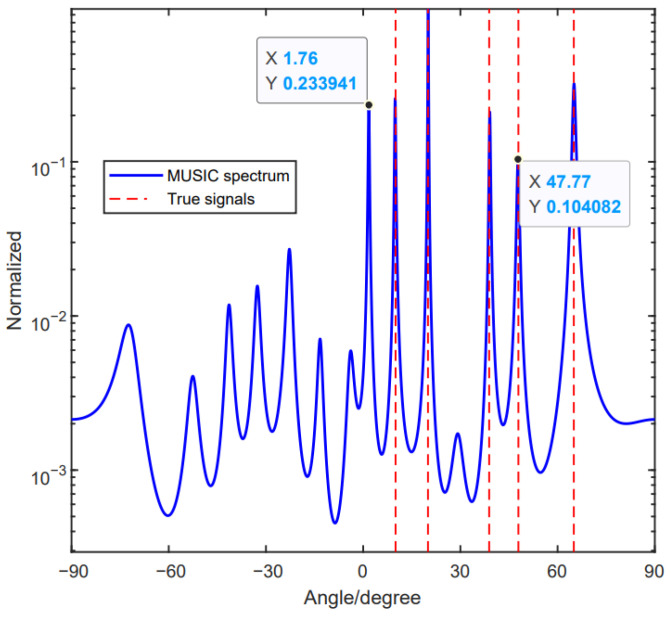
Using the method in [34] for the beamforming technique sometimes is not effective.

**Figure 7 sensors-24-06205-f007:**
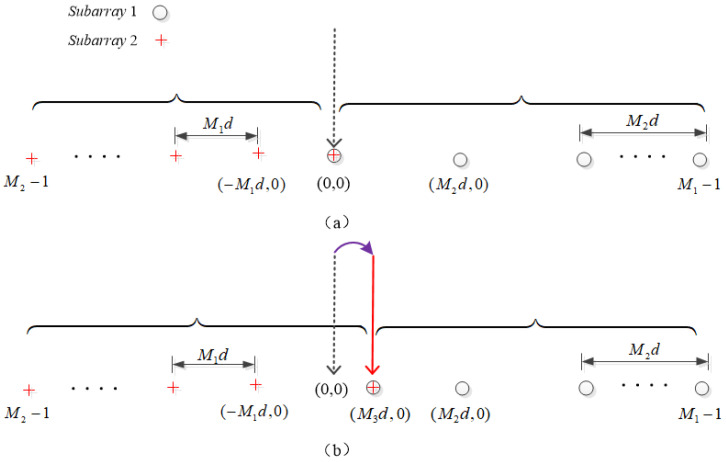
(**a**) The unfolded coprime linear array. (**b**) The designed and improved unfolded coprime linear array.

**Figure 8 sensors-24-06205-f008:**
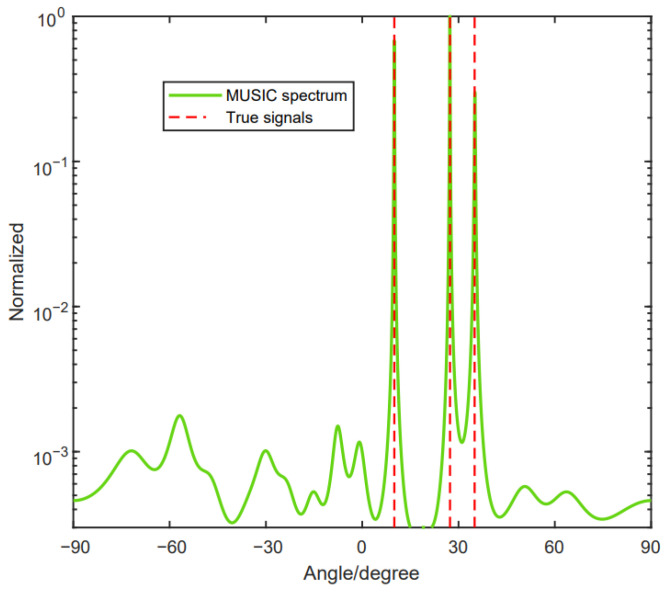
The reconstructed array configuration can achieve the full DOFs of three source signals.

**Figure 9 sensors-24-06205-f009:**
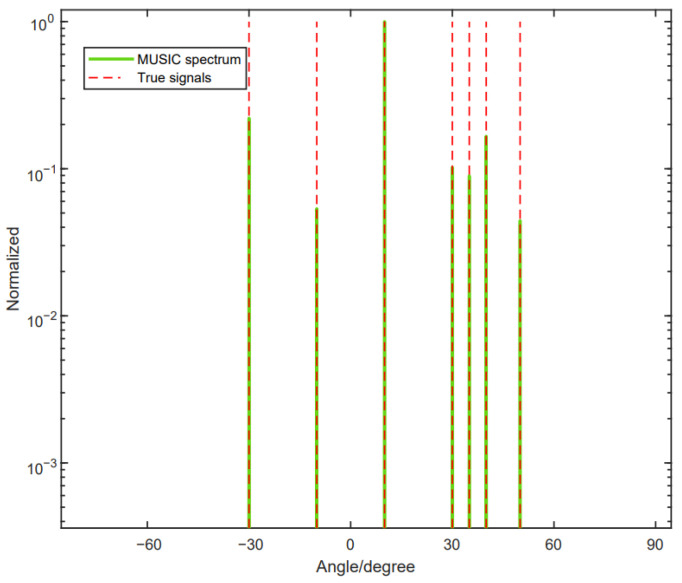
The reconstructed array configuration can achieve the full DOFs of seven source signals.

**Figure 10 sensors-24-06205-f010:**
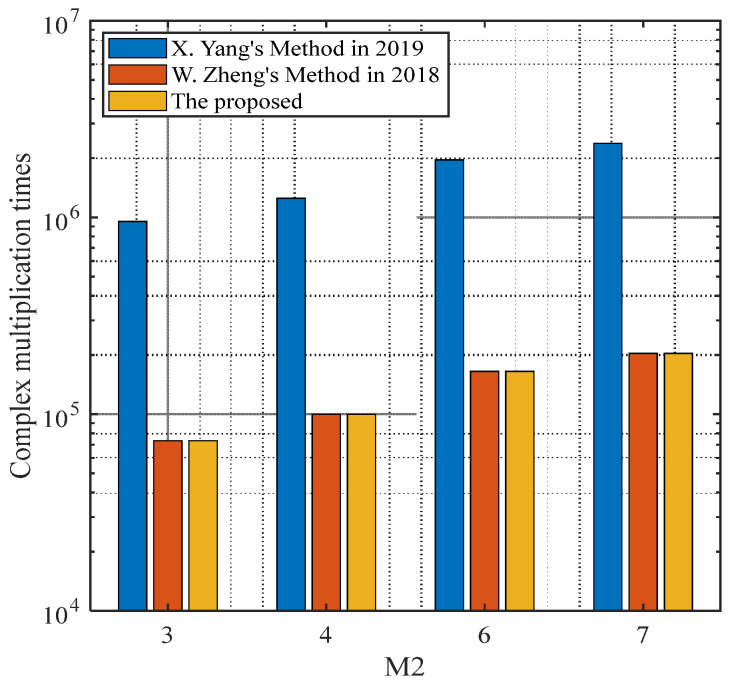
The complexity versus the number of sensors [33,34].

**Figure 11 sensors-24-06205-f011:**
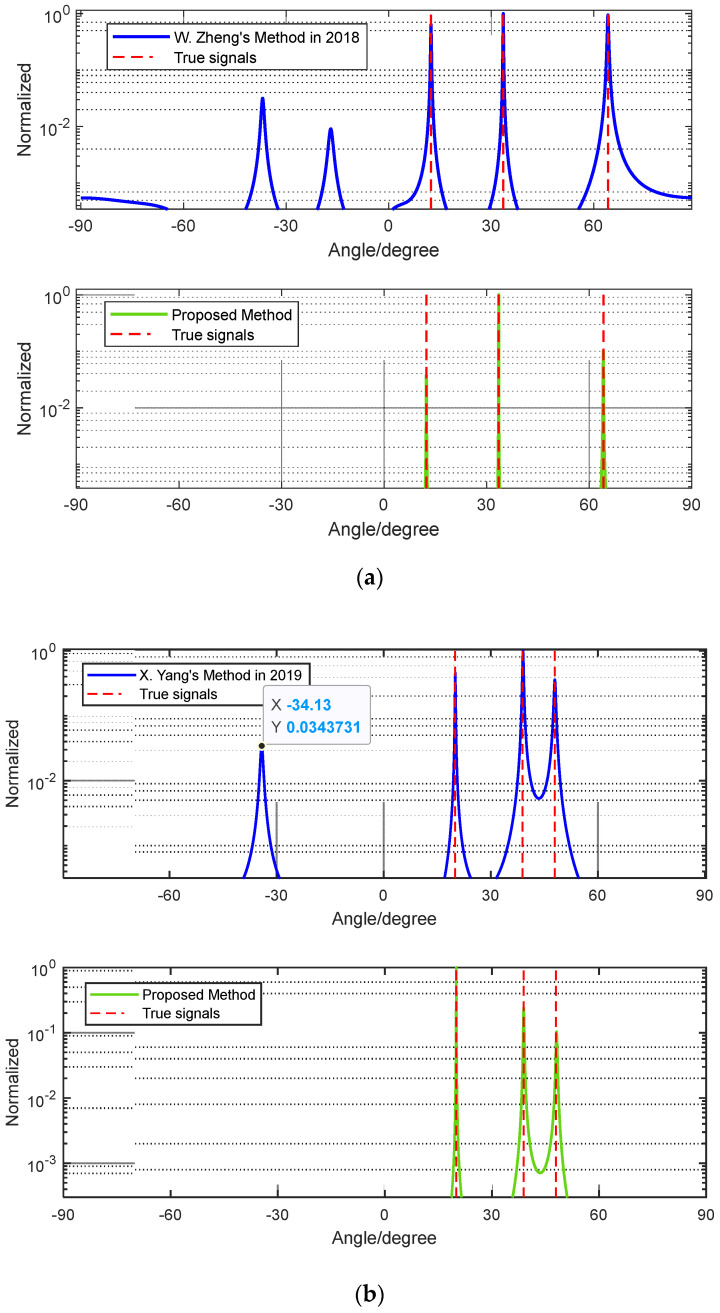
(**a**) Comparison of the proposed method to the method in [33] and (**b**) comparison of the proposed method to the method in [34].

**Figure 12 sensors-24-06205-f012:**
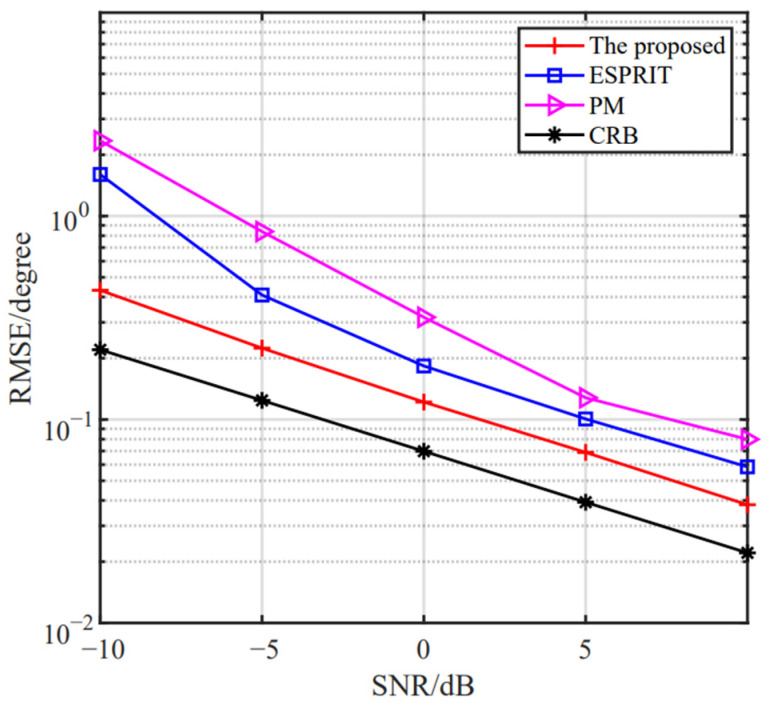
The RMSE versus the SNR of the proposed method.

**Figure 13 sensors-24-06205-f013:**
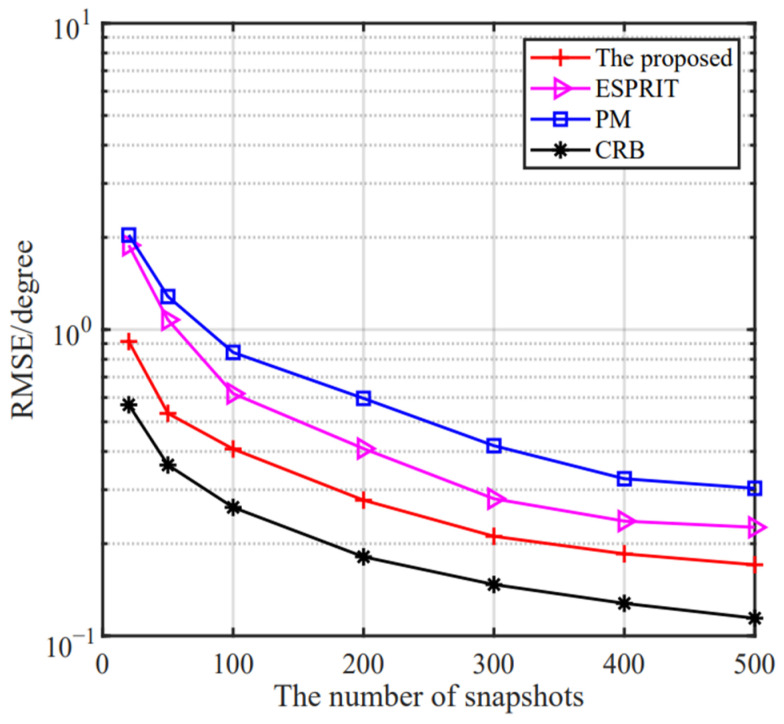
The RMSE versus the snapshot of the proposed method.

**Figure 14 sensors-24-06205-f014:**
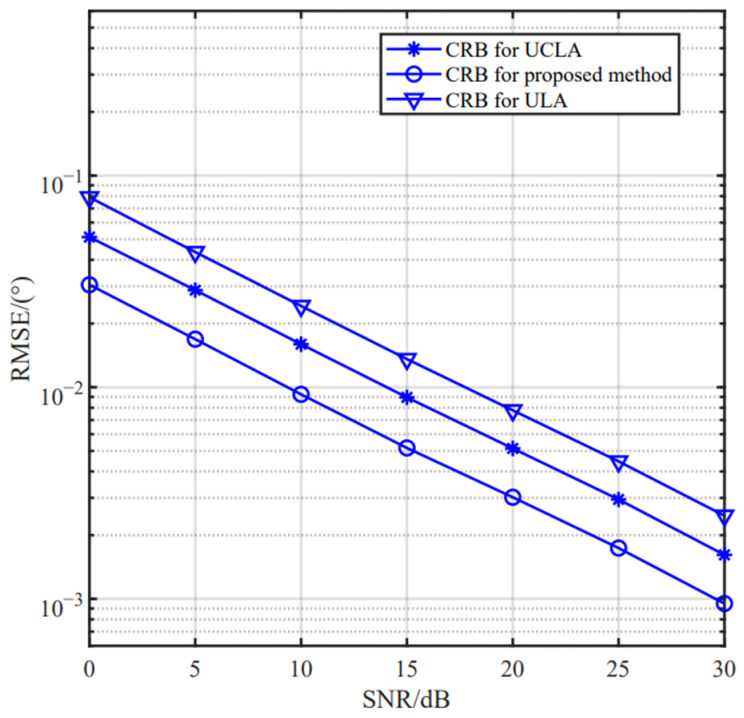
The RMSE versus the SNR based on different arrays.

**Figure 15 sensors-24-06205-f015:**
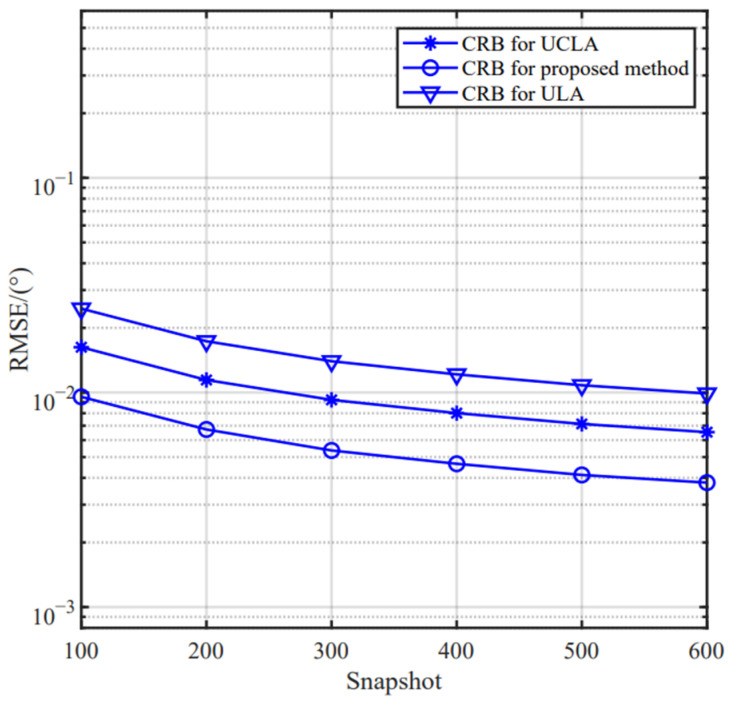
The RMSE versus the snapshot based on different arrays.

**Table 1 sensors-24-06205-t001:** Comparison of computational complexity.

Algorithms	Computational Complexity	Running Time
Method in [33]	$O(T^{2}L+T^{3}+GT(T-K))$	4.1106 ms
Method in [34]	$O(T^{2}L+T^{3}+GT(T-K)+QT^{2})$	6.0122 ms
The proposed	$O(T^{2}L+T^{3}+GT(T-K))$	4.1245 ms

## Data Availability

Data are contained within the article.

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
