# Peer review of "An Improved Unfolded Coprime Linear Array Design for DOA Estimation with No Phase Ambiguity"

_sensors, 2024, doi:10.3390/s24196205_

Round 1
Reviewer 1 Report (New Reviewer)
Comments and Suggestions for Authors
The authors present a novel UCLA strategy to solve the phase ambiguity problem. The reference element is rearranged and the reshaped directional property is used to eliminate the ambiguity problem. The aperture is exploited and the computational complexity is reduced. This work is interesting but I have the following concerns.
1. For the ambiguity problem, P. Pal has discussed in the sparse array condition that MUSIC method can be directly used. This can be found in the published paper "Direct-MUSIC on sparse arrays" and "Why does direct-MUSIC on sparse-arrays work?" I understand that the authors proposed a new array design in this work. But it seems quiet important to clarify the different between this work and the two above mentioned papers.
2. In Fig 8 and 9, authors mentioned full DOFs of 3 and 7 signals. How to obtain this full DOFs? Pls provide more details.
3. In simulations, only very few methos are compared, pls provide more comparisons with more methods.
Comments on the Quality of English LanguageThe English should be further improved. For instance, line 236-237, line335-336 and etc., pls check.
Author Response
Dear Reviewer:
Our manuscript entitled “An Improved Unfolded Co-prime Linear Array Design for DOA Estimation with No Phase Ambiguity”, was revised according to the reviewer's comments, and the itemized response to reviewer’s comments is attached. We deeply appreciate your consideration of our manuscript. If you have any queries, please don’t hesitate to contact me at the address below.
Thank you and best regards.
Yours sincerely,
Pan Gong
Corresponding author:
Name: Pan Gong
E-mail: gongpan@nuaa.edu.cn

Reviewer 2 Report (New Reviewer)
Comments and Suggestions for Authors
Authors propose an unfolded co-prime linear array design for direction of arrival estimation trying with elimination of phase ambiguity. The proposal shows some novelty and results could be promising, however, there are many issues to be addressed before publication. Some of my concerns that need to be addressed include:
1) Authors do not mention DoA estimation techniques that use neural networks in order to improve accuracy and reduce time of estimation. Authors should compare their proposed method with ML-based models, and refer to recently published proposals, such as:
C. M. Mylonakis et al., "3D Direction of Arrival Estimation: An Innovative Deep Neural Network Approach," 2024 13th International Conference on Modern Circuits and Systems Technologies (MOCAST), Sofia, Bulgaria, 2024, pp. 01-04, doi: 10.1109/MOCAST61810.2024.10615339.
2) On page 2, lines 65-66, the authors mention "Aiming to tackle the problem, Yang et. al. proposed a beamforming based technique by defining a decision variable [34] to eliminate the phase ambiguity". It is highly suggested that authors add a paragraph explaining whether their proposed method could potentially collaborate with other beamforming approaches, such as those aiming at decreasing the sidelobe level.
3) On lines 239-240, the authors claim "Consequently, two different steering vectors can be attained. By using this property, the angle ambiguity can be eliminated. ", but they neither explain why nor how angle ambiguity can be better supported and enhanced through their proposal, when compared to other similar proposals. A brief yet accurate explanation of previously presented theoretical advantages should be added and emphasized more on these lines.
4) On page 11, the running time should be defined more clearly.
Comments on the Quality of English LanguageThere are many syntax and grammar errors, which the authors should revise. For example, on page 4 of 17: "In the case 1" should be "In case 1", "θ2.Therefore" should be "θ2. Therefore", on line 164 "consider three" should be "we consider three", and many more.
Author Response
Dear Reviewer:
Our manuscript entitled “An Improved Unfolded Co-prime Linear Array Design for DOA Estimation with No Phase Ambiguity”, was revised according to the reviewer's comments, and the itemized response to reviewer’s comments is attached. We deeply appreciate your consideration of our manuscript. If you have any queries, please don’t hesitate to contact me at the address below.
Thank you and best regards.
Yours sincerely,
Pan Gong
Corresponding author:
Name: Pan Gong
E-mail: gongpan@nuaa.edu.cn

This manuscript is a resubmission of an earlier submission. The following is a list of the peer review reports and author responses from that submission.
Round 1
Reviewer 1 Report
Comments and Suggestions for Authors
Aiming at the direction of arrival estimation problem, this paper researches the phase ambiguity for unfolded coprime linear array. Meanwhile, a lower bound for unbiased estimation based on CRLB is provided. The overall structure of the paper is good, but there are the following issues:
(1) The format of the paper needs to be carefully checked, and some formulas are not in the appropriate positions. For example, the formulas after "As dissected in Figure 1, we consider an UCLA with" have been moved up. Suggest making an overall adjustment.
(2) For angle estimation problems, there are conventional beamforming, minimum variance distortion free response, and traditional methods such as subspace decomposition based multi signal classification used in this paper, as well as signal estimation using deep learning in recent years. The author needs to be more comprehensive in analyzing the current situation both domestically and internationally.
(3) The biggest problem with this article is that the comparative experimental section is too simplistic, relying solely on true values or evaluations based on CRLB, which I believe is far from enough. Suggest the author to add comparative algorithms.
Author Response
Revision Report
Dear Reviewer
Thank you for your Email, and many thanks for coordination with our paper.
First of all, the authors would like to thank the associate editor and the reviewers for the time and effort they have put into our paper. We have carefully read all of your comments and have made necessary modifications to our revised manuscript, in which the modified portions are highlighted in blue. We believe the presentation of the paper has been improved.
My manuscript (ID-2764088) has been revised and refreshed by our co-authors. According to the reviewers’ comments, I hereby write a revision report and list the following modifications. Please refer to the attachment for specific modifications.

Reviewer 2 Report
Comments and Suggestions for Authors
- The abstract misses the normal parts and is hard to correctly interpret.
- Lay-out of the text is very poor. A lot of superscript is used in a wrong way and makes it very difficult to understand.
- The first part of the paper is very basic and is a textbook analysis of the phase ambiguity problem.
- The text is extremely chaotic, contains a lot of typing errors (combing, ger, sinals, case2, MUSCI…) and many sentences are very unclear (missing subject, too long, missing verb or adverb). For instance: “Coprime array, which is made up of two uniform linear arrays and the inter element spacing is larger than half wavelength.” and “Since the element spacing is larger than half wavelength, the phase ambiguity problem results in.”
- Are you sure that the inter element spacing is d1 = M2 d?
- Please explain the calculations and formulas used in Figure 2.
- Please elaborate the CRB by explaining the formula and practical use in your theory.
- This paper is based on another paper, which is even not referred. The incremental extra is limited and hence not worth the be published.
- For the Monte Carlo simulation you set P = 1000. Please prove that this number is high enough.
- The method in [34] seems to have the same complexity as yours. Can you prove that your method is better than the one in the literature? What about the CRB for [33] and [34]? Can you compare with them?
Comments on the Quality of English Language- The abstract misses the normal parts and is hard to correctly interpret.
- Lay-out of the text is very poor. A lot of superscript is used in a wrong way and makes it very difficult to understand.
- The first part of the paper is very basic and is a textbook analysis of the phase ambiguity problem.
- The text is extremely chaotic, contains a lot of typing errors (combing, ger, sinals, case2, MUSCI…) and many sentences are very unclear (missing subject, too long, missing verb or adverb). For instance: “Coprime array, which is made up of two uniform linear arrays and the inter element spacing is larger than half wavelength.” and “Since the element spacing is larger than half wavelength, the phase ambiguity problem results in.”
- Are you sure that the inter element spacing is d1 = M2 d?
- Please explain the calculations and formulas used in Figure 2.
- Please elaborate the CRB by explaining the formula and practical use in your theory.
- This paper is based on another paper, which is even not referred. The incremental extra is limited and hence not worth the be published.
- For the Monte Carlo simulation you set P = 1000. Please prove that this number is high enough.
- The method in [34] seems to have the same complexity as yours. Can you prove that your method is better than the one in the literature? What about the CRB for [33] and [34]? Can you compare with them?
Author Response
Dear Reviewer
Thank you for your Email, and many thanks for coordination with our paper.
First of all, the authors would like to thank the associate editor and the reviewers for the time and effort they have put into our paper. We have carefully read all of your comments and have made necessary modifications to our revised manuscript, in which the modified portions are highlighted in blue. We believe the presentation of the paper has been improved.
My manuscript (ID-2764088) has been revised and refreshed by our co-authors. According to the reviewers’ comments, I hereby write a revision report and list the following modifications. Please refer to the attachment for specific modifications.
